# Antarctic Basal Water Storage Variation Inferred from Multi-Source Satellite Observation and Relevant Models

**Jingyu Kang** [1,2], **Yang Lu** [1,2], **Yan Li** [1,2], **Zizhan Zhang** [1,2] **and Hongling Shi** [1,2,*]

[1] State Key Laboratory of Geodesy and Earth's Dynamics, Innovation Academy for Precision Measurement Science and Technology, Wuhan 430071, China; kangjingyu17@mails.ucas.ac.cn (J.K.); luyang@asch.whigg.ac.cn (Y.L.); liyan@asch.whigg.ac.cn (Y.L.); zzhang@asch.whigg.ac.cn (Z.Z.)

[2] University of Chinese Academy of Sciences, Beijing 100049, China

\* Correspondence: hlshi@asch.whigg.ac.cn

**Abstract:** Antarctic basal water storage variation (BWSV) refers to mass changes of basal water beneath the Antarctic ice sheet (AIS). Identifying these variations is critical for understanding Antarctic basal hydrology variations and basal heat conduction, yet they are rarely accessible due to a lack of direct observation. This paper proposes a layered gravity density forward/inversion iteration method to investigate Antarctic BWSV based on multi-source satellite observations and relevant models. During 2003–2009, BWSV increased at an average rate of $43 \pm 23$ Gt/yr, which accounts for 29% of the previously documented total mass loss rate ($-76 \pm 20$ Gt/yr) of AIS. Major uncertainty arises from satellite gravimetry, satellite altimetry, the glacial isostatic adjustment (GIA) model, and the modelled basal melting rate. We find that increases in basal water mainly occurred in regions with widespread active subglacial lakes, such as the Rockefeller Plateau, Siple Coast, Institute Ice Stream regions, and marginal regions of East Antarctic Ice Sheet (EAIS), which indicates the increased water storage in these active subglacial lakes, despite the frequent water drainage events. The Amundsen Sea coast experienced a significant loss during the same period, which is attributed to the basal meltwater discharging into the Amundsen Sea through basal channels.

**Keywords:** Antarctica; basal water variation; multi-source satellite; gravity inversion

## 1. Introduction

In Antarctica, downward overburden from overlying thick ice and heat supplied by geothermal and interfacial friction have generated a large amount of meltwater on the interface between the ice sheet and its underlying bed [1,2]. Typically, the basal meltwater could spread across the ice-bed interface, accumulate in subglacial lakes, migrate through complex basal hydrological networks, or flow into surrounding oceans [2–4]. The presence of basal water storage facilitates fast ice flow by lubricating the interface between the ice sheet base and bed materials, and the variation in basal water storage may have an effect on basal effective friction and trigger changing ice velocity [5–7].

Antarctic basal water storage variations (BWSV) are controlled by basal conditions (basal temperature, geothermal flux, melting, freezing, etc.) and basal mass balance (BMB). Basal conditions affect BWSV by controlling basal ice melting/ water freezing, and have been studied by many researchers [8–17], through regional or continental radioglaciological, geological, magnetic, seismic, sparse ice-core site measurements data, or ice-sheet models. BMB affects BWSV through basal water migrations on the ice-bed interface, causing the ice sheet's vertical movement (IVM) through pumping up and down the overlying ice sheet [2,18,19]. However, the continental BMB remains poorly understood due to the lack of direct observation. Furthermore, changes in groundwater in basal aquifers also influence BWSV, while the current knowledge on such changes is mainly confined to coastal or ice-free regions of Antarctica [20,21] with a magnitude smaller than that of basal

water migrations [20]. Therefore, the induvial effect of groundwater changes on BWSV are ignored in this study.

To date, many studies have been conducted for the detection of BMB. The commonly adopted approach is to examine local IVM (surface height variations) by satellite altimetry, based on surface height variations in response to the basal water volume changes caused by subglacial lakes' filling or drainage [3,22]. However, this approach is effective only in regions that are characterized by periodic or abrupt basal water migrations and might be invalid in regions where basal water increase/decrease constantly, or where surface expressions of basal water migrations are not sufficient (such as the Siple Coast region [23]). Göeller et al. [24] proposed a balanced water layer concept on a continental scale to present the evolution of subglacial lakes or water fluxes, and demonstrated the variations of subglacial lakes and their effect on overlaying ice velocity, while the presented results rely largely on the reliability of the building models. Besides, many combined methods have also been conducted on AIS [25–29], with the object of reconciling the results among individual geodetic techniques (including satellite altimetry, satellite gravimetry, and GPS) by adjusting the proportion of mass change processes of AIS (such as surface mass balance, ice dynamics, and GIA). Unfortunately, these methods are unable to isolate BMB due to the fact that the Earth's surface 'thin layer' assumption used in gravity-mass conversion process [30,31] ignores dual sensitivities of gravity variations to mass and distance variations [32]. To address this problem, gravity forward/inversion methods [33] can be used to exploit the dual sensitivities of gravity and convert the known mass variations to gravity variations and vice versa. In this way, BMB-induced gravity variations can be obtained by subtracting variations caused by other components from the total Antarctic gravity variations, and the associated mass variations can also be estimated accordingly.

To investigate the BMB and BWSV, a layered gravity density forward/inversion iteration method is presented. Unlike previous direct/indirect local observation and numerical simulation approaches, the proposed method relies little on conceptual ice models, but on the input datasets consisting of multi-source satellite geodetic observation and relevant models throughout AIS. Thus, the BMB and BWSV results can cover the most regions of AIS. Uncertainties in BMB and BWSV depend mainly on the errors of the input datasets rather than the reliability of conceptual modes. The proposed method provides a new approach for exploring the mass variation beneath the AIS, which is important for understanding the Antarctic mass changes and the evolution of Antarctic basal conditions.

## 2. Materials and Methods

### 2.1. Overview of Mass Redistributions of AIS

Active mass redistributions of AIS, from surface to solid Earth, are dominated by the following processes: snow accumulation, sublimation/runoff, and melting/firn compaction in firn layer (FL), ice dynamic flows (IDF), and ice sheet vertical movement (IVM) in ice layer, BMB and basal melting/freezing in the ice-bed interface, glacial isostatic adjustment (GIA) of solid Earth (Figure 1). These mass redistributions processes, according to the resulting impact on mass or height variations, can be classified into two categories: 1. mass-changing processes that contribute to mass variations, including snow accumulation, sublimation/runoff, IDF, BMB, and GIA; 2. mass-conserving processes that contribute to height variations without mass variations, including melting, firn compaction and IVM. It is notable that IVM contains not only the ice layer's vertical movement caused by basal meltwater migrations (or BMB), but also the movement caused by ice-water volume changes due to basal melting/freezing.

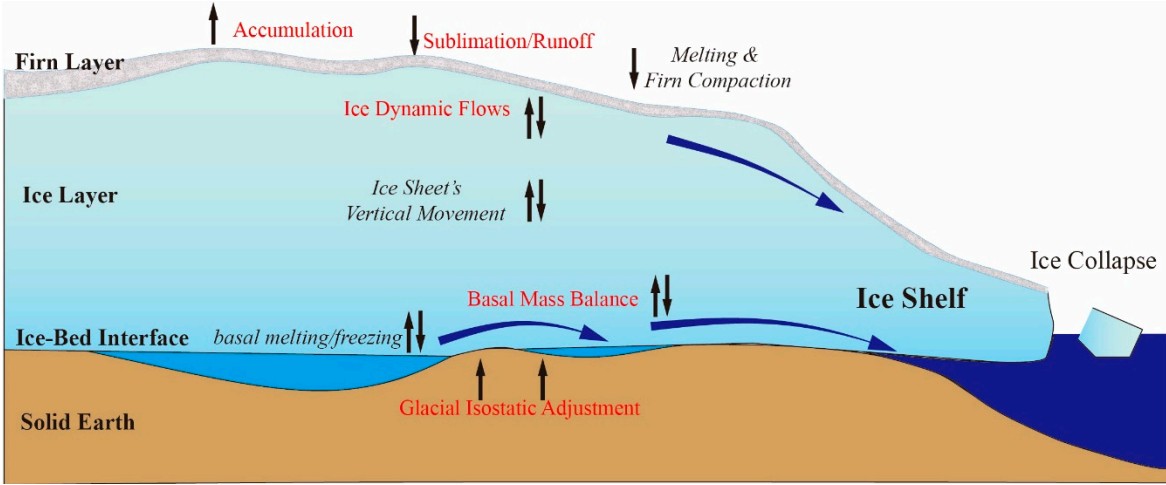

**Figure 1.** Mass redistributions in AIS. Red texts are components that contribute to mass-changing processes, and italic texts are components related to mass-conserving processes. Vertical arrows next to the texts denote the contribution to height variation.

Accordingly, mass redistributions of AIS are expressed as superpositions of gravity/height variations in each layer:

$$dg_{Ant} = dg_{FL} + dg_{IDF} + dg_{IVM} + dg_{BMB} + dg_{GIA} \tag{1}$$

$$dh_{Ant} = dh_{FL} + dh_{IDF} + dh_{IVM} + dh_{GIA} \tag{2}$$

where $dg_{Ant}$, $dh_{Ant}$ are exterior gravity variations and surface height variations of AIS (observed by GRACE and ICESat, respectively; see Sections 2.4.1 and 2.4.2), respectively. Other subscripts of the terms in Equations (1) and (2) represent height and gravity variations caused by associated mass redistribution processes. Compared with the terms in Equation (1), the BMB-related term is missing from Equation (2) because the BMB-induced height variations have already been contained in the $dh_{IVM}$ term.

*2.2. Estimation of BMB and BWSV*

Separating BMB from the total mass variations is performed based on different sensitivities of satellite observations on gravity and height variations of AIS. Then, BMB-induced gravity variations $dg_{BMB}$ can be expressed through the modification of Equation (1):

$$dg_{BMB} = dg_{Ant} - dg_{FL} - dg_{IDF} - dg_{IVM} - dg_{GIA} \tag{3}$$

In Equation (3), $dg_{Ant}$ and $dg_{GIA}$ are available from GRACE and GIA (see Section 2.4.3). The remaining terms ($dg_{FL}$, $dg_{IDF}$, and $dg_{IVM}$) are gravity variations related to associated height variations ($dh_{FL}$, $dh_{IDF}$, and $dh_{IVM}$) in Equation (2). However, satellite altimetry's limitation in vertical resolution makes it challenging to determine each height variation in Equation (2). To address this problem, we adopted a layered gravity density forward/inversion iteration method to separate $dg_{BMB}$ and estimate BMB based on com satellite altimetry/gravity data. Detailed procedures of the iteration method are described as follows.

2.2.1. Initialization of the Iteration Procedure

Iterative procedure initiates with the assumption that no basal water migrations (no BMB) occur in AIS. Accordingly, the initial value of $dh_{IVM}$ and $dg_{IVM}$ in Equations (2) and (3) are set to 0. To ensure the time consistency of the input data, the $dh_{GIA}$ term in Equation (2), derived from GIA model-predicted height variation, is constrained by spare GPS observations during the study period (see Section 2.4.3). Second, $dh_{GIA}$ and $dh_{IVM}$ are deducted from $dh_{Ant}$ to obtain the initial residual height variation $dh_{RHV}$ (that is, $dh_{RHV} = dh_{Ant} - dh_{GIA} - dh_{IVM}$).

It is worth noting that $dh_{RHV}$ only contains height variations caused by the firn layer's processes and ice dynamic flow. Third, we utilize the density discrimination method, simplified from the approach of Gunter et al. [25], to separate $dh_{FL}$, $dh_{IDF}$ from $dh_{RHV}$ and assign corresponding densities. The simplified density discrimination method differs from Gunter et al. [25] in that it relies primarily on satellite altimetry observation data rather than modelled climate data. The simplified density discrimination method is described as follows:

$$\rho = \begin{cases} \rho_{firn} \text{ for } dh_{FL}, \ \rho_{ice} \text{ for } (dh_{RHV} - dh_{FL}) & \text{if } (dh_{RHV} - dh_{FL}) < 0 \ \& \ |dh_{RHV} - dh_{FL}| > 2\sigma_{dh} \\ \rho_{firn} \text{ for } dh_{RHV} & \text{otherwise} \end{cases} \quad (4)$$

where $\rho_{firn}$ is firn density distribution of AIS [34]; $dh_{FL}$ is height variation caused by the spatio-temporal evolution of the firn layer and available from a time-dependent firn densification model (FDM) (see Section 2.4.4). $\rho_{ice}$ is ice layer density (917 kg m$^{-3}$). The resulting uncertainty is expressed as $\sigma_{dh} = \sqrt{\delta_{Ant}^2 + \delta_{FL}^2}$.

As shown in Equation (4), the negative height differences between $dh_{RHV}$ and $dh_{FL}$ greater than $2\sigma_{dh}$ are assumed to be caused by IDF (that is, $dh_{IDF} = dh_{RHV} - dh_{FL}$), then $\rho_{ice}$ and $\rho_{firn}$ are assigned to $dh_{IDF}$ and $dh_{FL}$ respectively. In other cases, it is assumed that no IDF occurs. Therefore, $\rho_{firn}$ is assigned to $dh_{RHV}$. Based on the assumption above, the associated gravity variations ($dg_{IDF}$, $dg_{FL}$) are estimated through the gravity forward modelling method (see Supplementary Materials).

### 2.2.2. Estimating BMB and BWSV through Iteration Method

Based on the initial value of $dg_{FL}$, $dg_{IDF}$, and $dg_{IVM}$, we calculate $dg_{BMB}$ according to Equation (3) and estimate initial BMB ($m_{BMB}$, expressed by equivalent water height, EWH) using a layered gravity density inversion method (see Supplementary Materials). Then, a 300 km gaussian smoothing is applied on $m_{BMB}$, in order to match the spatial resolution of GRACE. It is worth noting that the layered gravity density inversion method is applied based on the assumption that basal water migrations occur within a thin layer in the ice-bed interface. In this process, BMB is estimated by solving the inversion problem of the thin layer's density. However, the initial BMB result is 'unrealistic' because the thin layer assumption ignores the volume variations caused by basal water migrations as water is nearly incompressible. The increase/decrease in basal water caused by basal water migration would lead to the lift/drop of the overlying ice [2,18]. Following the same logic, the basal ice-water volume changes induced by basal melting/freezing could result in the ice sheet's lift/drop as well. These ice sheet's lift/drop processes, referred to as ice sheet movement (IVM) in this paper, are also included in surface height variations ($dh_{IVM}$ in Equation (2)), and will in turn affect BMB estimation. To solve this problem, we developed an iteration method described as follows. First, we combined initial EWH of BMB and modelled basal melting rate data ($m_{BM}$, see Section 2.4.4) to estimate $dh_{IVM}$: in basal melting region (where $m_{BM} > 0$), BMB is expressed in the form of liquid water, and the associated $dh_{IVM}$ is expressed by $dh_{IVM} = m_{BMB}$; in basal freezing region (where $m_{BM} = 0$), BMB is experienced in the form of ice, and the associated $dh_{IVM}$ is expressed by $dh_{IVM} = m_{BMB} * \rho_{ice}/1000$. Second, $dh_{IVM}$ is used to recalculate the IVM-induced gravity variations $dg_{IVM}$. This process can be simplified by only updating the $dh_{IVM}$ term in Section 2.2.1. Third, the subsequent procedures in Sections 2.2.1 and 2.2.2 are implemented iteratively, until the $m_{BMB}$ is stable (the total BMB differences between two consecutive iterations is smaller than 5 Gt/yr). Finally, BWSV is estimated through the combination of $m_{BMB}$ and $m_{BM}$: in basal melting regions, BWSV is mainly subject to liquid basal water migrations and the melting of the ice base, and can be expressed by $m_{BWSV} = m_{BMB} + m_{BM}$; in basal freezing regions, $m_{BWSV}$ is 0 because liquid water would not occur in these regions. The flowchart of the iterative procedure is shown in Figure 2.

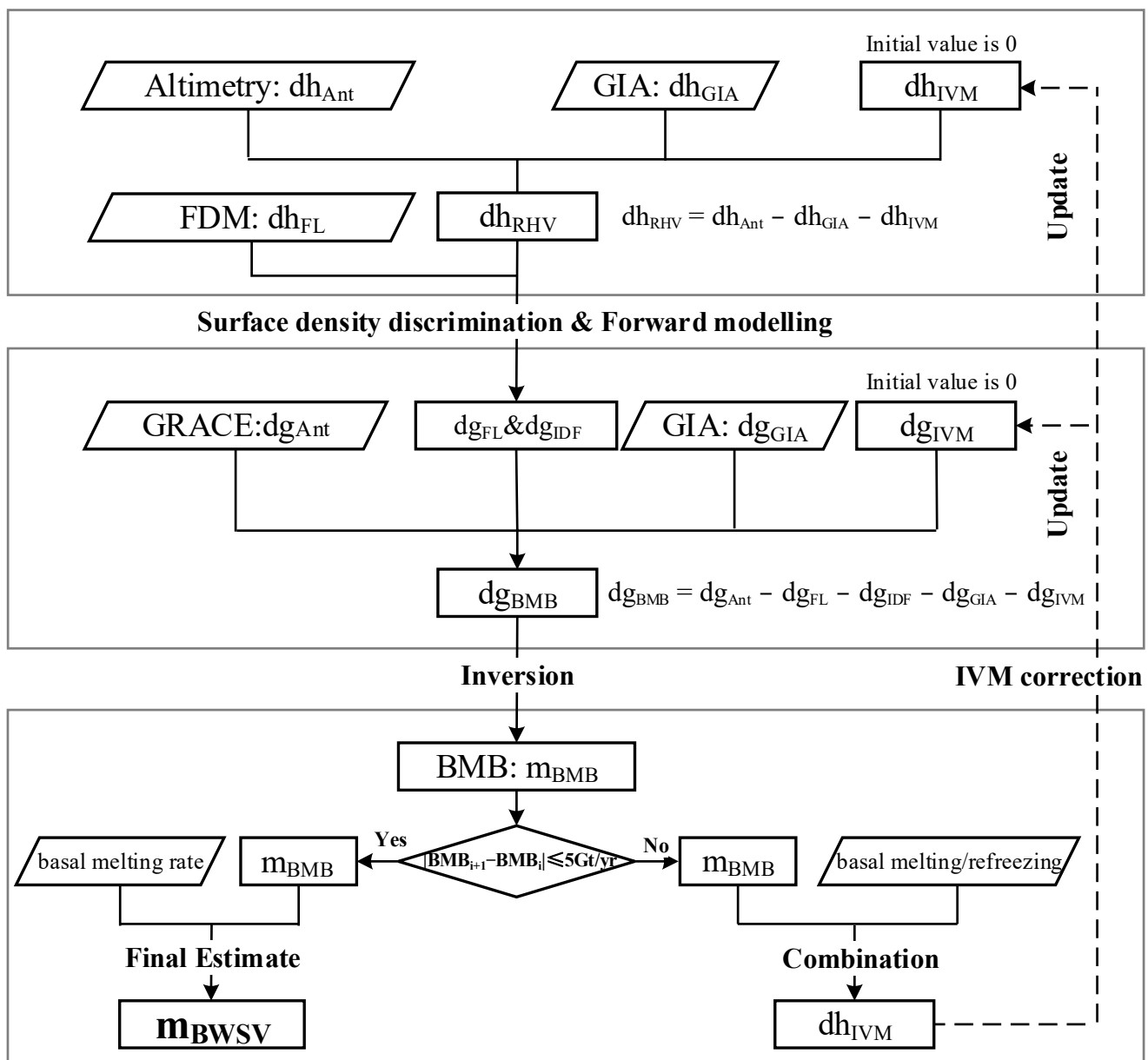

**Figure 2.** Flowchart for estimating BMB and BWSV. In upper pane, initial values of $dg_{FL}$ and $dg_{IDF}$ are calculated through surface density discrimination and gravity forward modelling method. In middle pane, $dg_{BMB}$ are abstracted from the total gravity variations $dg_{Ant}$, and be used to calculate $m_{BMB}$ through layered gravity inversion method. In lower pane, $m_{BMB}$ and basal melting data are combined to obtain $dh_{IVM}$, then the obtained $dh_{IVM}$ are used to update the relevant variations in the first step for iteration until the BMB result is stable.

### 2.3. Uncertainty Estimation

Uncertainty estimation is performed according to the following three steps. In step 1, uncertainty of each input dataset is converted to corresponding gravity uncertainties (terms in the right side of Equation (5)). Among them, $\delta dg_{Gravi}$ is derived from the calibrated errors of spherical harmonic coefficients provided by CSR, JPL, GFZ, and errors of degree 1&2 (see Section 2.4.1), through the method that modified from Wahr et al. [35]; $\delta dg_{Alti}$ is derived from the standard deviations of height variations that relate to satellite altimetry and inter-campaign biases corrections (see Section 2.4.2) and corresponding firn density, through the error propagation law of the gravity reduction of cylinder model [32]. Similarly, $\delta dg_{FDM}$, $\delta dg_{GIA}$, $\delta dg_{GPS}$ are derived height variations errors related to FDM, GIA models

and GPS observations (see Sections 2.4.3 and 2.4.4), through the method above mentioned. In step 2, gravity uncertainty of BMB ($\delta dg_{BMB}$) is estimated through Equation (5). This uncertainty estimation method does not strictly follow the procedure in Section 2.2 because the uncertainty of the iteration procedure itself is difficult to assess.

$$\delta dg_{BMB} = \sqrt{\delta dg_{Gravi}^2 + \delta dg_{Alti}^2 + \delta dg_{FDM}^2 + \delta dg_{GIA}^2 + \delta dg_{GPS}^2} \tag{5}$$

In step 3, mass uncertainty of BMB ($\delta m_{BMB}$) is derived from $\delta dg_{BMB}$, through the inversion of the error propagation law of the gravity reduction of the cylinder model. Then, $\delta m_{BWSV}$ is estimated as follows:

$$\delta m_{BWSV} = \sqrt{\delta m_{BMB}^2 + \delta m_{BM}^2} \tag{6}$$

where $\delta m_{BM}$ is the standard deviation of basal melting rate.

### 2.4. Input Data Processing

#### 2.4.1. Gravimetry

Exterior time-varying gravity variations of AIS ($dg_{Ant}$ in Equations (1) and (3)) are available from three Release 06 (RL06) monthly GRACE gravity field solutions provided by CSR, JPL, and GFZ. Each of the solutions is represented by fully normalized Stokes potential coefficients with degree and order up to 60. In pre-processing process, the degree one coefficients are added to of each GRACE solutions using values generated from the approach of Swenson et al. [36]. The C20 coefficients are replaced by the values derived from satellite laser ranging [37]. Striping errors are suppressed by P4M6 smoothing [38] and 300 km Gaussian smoothing together. The leakage-out errors and amplitude dampening are restored by multiplying a scaling factor [39]. It is known that the BMB only occurs in AIS; therefore, leakage-in errors corrections were not performed in this study. Then, the linear term was abstracted by utilizing a least-squares adjustment function containing four parameters (constant term, linear term, and annual periodic term). The linear term of AIS gravity variations $dg_{Ant}$ is expressed as follows:

$$dg_{Ant} = \frac{GM}{R^2} \sum_{n=2}^{max} (n-1) \frac{1}{1+k_n} \sum_{m=0}^{n} \overline{P}_{nm}(\cos(\theta))[\Delta C_{nm}\cos(m\phi) + \Delta S_{nm}\sin(m\phi)] \tag{7}$$

where GM is the geocentric gravitational constant, R is the mean Earth radius, $k_n$ is the load Love number of degree n [30], $\overline{P}_{nm}$ are the normalized associated Legendre functions, $\Delta C_{nm}, \Delta S_{nm}$ are modified harmonic coefficients, and $\theta$ and $\phi$ are the colatitude and longitude of the gravity point, respectively. Equation (7) is derived from the fundamental equation of physical geodesy [40]; therefore, the obtained gravity variation $dg_{Ant}$ can be considered as the free-space anomaly variations in geoid. Correspondingly, the gravity forward/inversion in this study can also be considered as the removal of the variation of terrain correction from the free-space anomaly variations and its reverse application.

The uncertainties in the gravity variations are estimated by utilizing the method of Wahr et al. [35]. The calibrated errors of the harmonic coefficients are available from CSR, JPL, and GFZ, respectively; the coefficient errors of degree 1&2 are replaced by the associated standard deviations of Cheng and Swenson [36,37]. Figure 3a–f shows the linear gravity variations trend over AIS and corresponding uncertainties. The observation period is consistent with the satellite altimetry simultaneous observation period, in order to avoid sampling error.

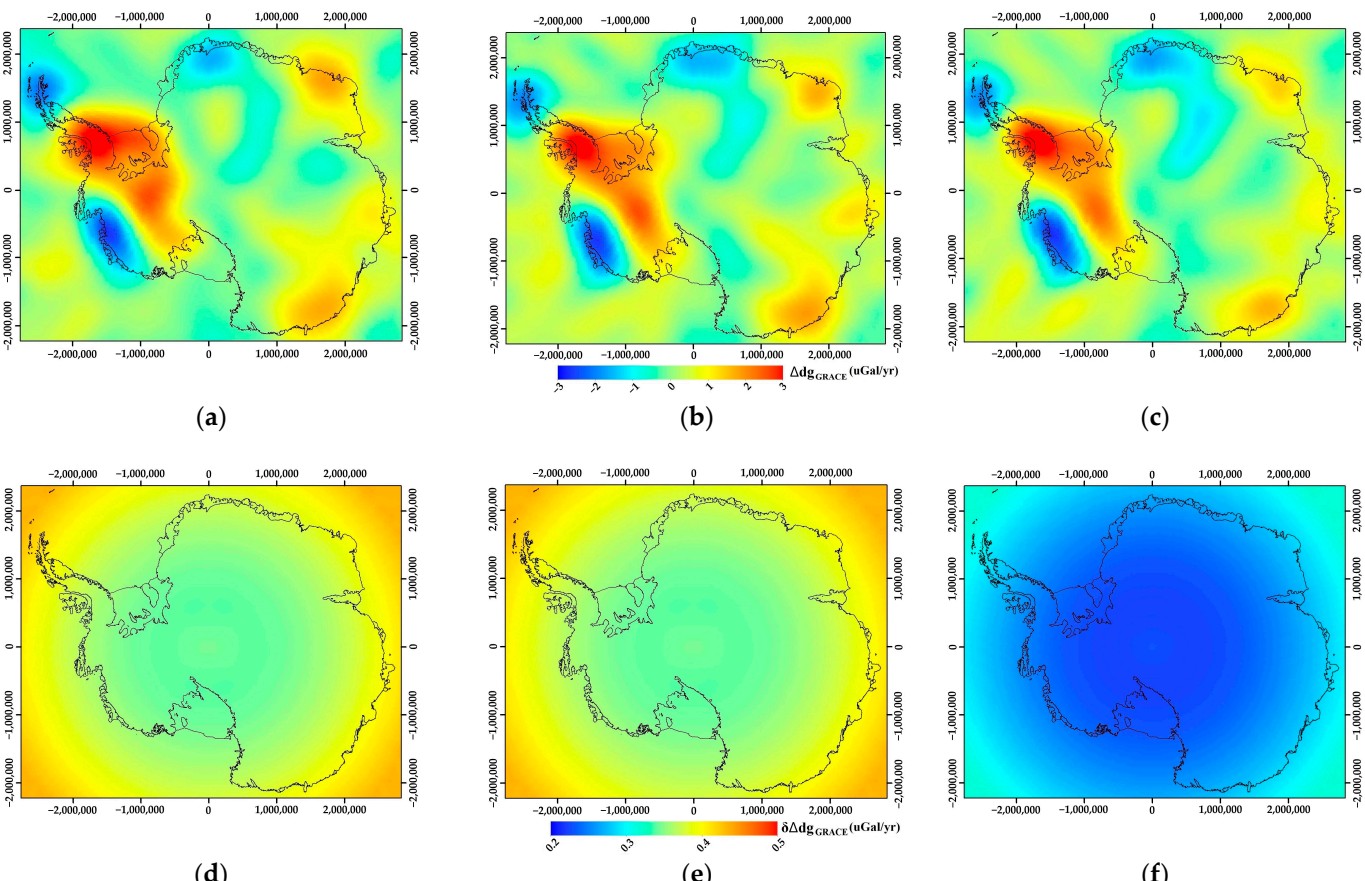

**Figure 3.** Linear gravity variation trend over AIS estimated from (**a**) CSR (**b**) JPL and (**c**) GFZ, and associated uncertainties (**d**–**f**). The observation period is consistent with the satellite altimetry simultaneous observation period.

### 2.4.2. Altimetry

Surface height variations of AIS are derived ICESat observations spanning from February 2003 to October 2009. The object of ICESat is to measure height variations of ice sheet with the accuracy $\leq 1.5$ cm/yr at the spatial resolution of $100 \times 100$ km$^2$ on AIS [41]. Accordingly, we utilized a block crossover analysis method [42,43] to estimate AIS height variations, and the size of the block was set to $100 \times 100$ km$^2$. In pre-processing, the crossover points with height variations greater than 10 m/yr are deleted in order to eliminate errors that arise from the small-scale surface roughness, undetected forward scattering, or interpolations between successive footprints [44]. Afterward, a 3σ criterion test is performed in each block to reduce the residual errors [45]. The linear surface height variations trends are substracted by utilizing the least-square adjustment method in Section 2.4.1.

The ICESat inter-campaign biases (ICB, the different biases from one ICESat campaign to the next one) have important effects on the long-term elevation change rate, and have been estimated in several studies. For example, Zwally et al. [46] computed the ICB by using concurrent radar altimetry on the same surface in open water and thin ice in leads and polynyas in Antarctic sea ice pack. Richter et al. [47] and Schroeder et al. [48] detected the ICB based on the near-zero surface height changes and hydrostatic equilibrium for the snow surface above Lake Vostok and its surroundings in East Antarctica (EA). Other studies estimated the ICB through the assumption of near-zero elevation changes regions in EA. However, these ICB results vary largely due to utilizing different areas and calibration methods across the globe, and none of them has been endorsed by the ICESat Science Team, NASA, or NSIDC, which leads to additional uncertainty in estimating height variations of

AIS. To reduce the uncertainty caused by ICB, we utilize the average ICB correction from Zwally et al. [46], Richter et al. [47] and Schroeder et al. [48], to avoid the artificial selection of different areas and calibration methods.

Figure 4 shows the linear surface height variations trend over AIS and associated uncertainties, with a grid size of $100 \times 100$ km$^2$. In order to match with the GRACE spatial resolution, a 300 km Gaussian smoothing filter is applied to the surface height variations before gravity forward modelling calculation.

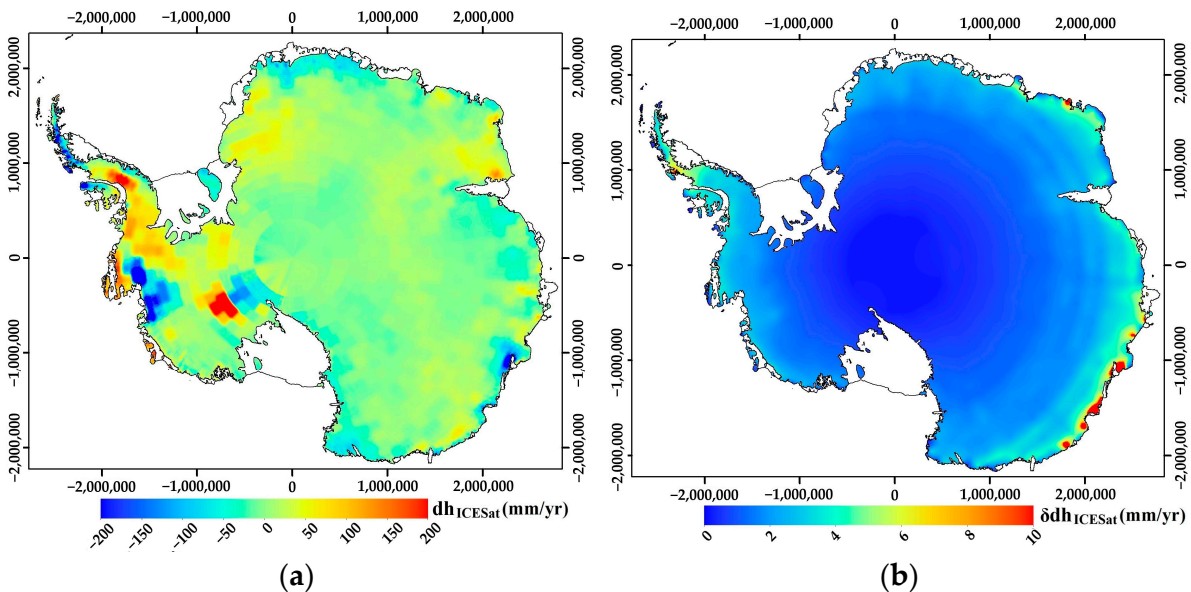

**Figure 4.** (**a**) Linear surface height variations trend derived from ICESat (including ICB correction) during February 2003–October 2009 (truncated at 200 mm/y). (**b**) Associated uncertainties.

### 2.4.3. GIA Models and GPS Data

Uncertainty in GIA has been considered as an important error source in evaluating the mass redistribution process. Many researchers have been worked on modelling GIA through various methods [49]. In this study, three GIA models (ICE-6G, IJ05_R2, and W12a) are used to account for the secular deformation of solid Earth [50–53]. Among them, ICE-6G is global GIA model that is constrained by geological and geodetic observation data including GPS, ice thickness, relative sea level histories, and the age of marine sedimentation. The associated uncertainty is estimated to be 0.89 mm/y according to the difference between the uplift rate derived from ICE-6G and that observed by 42 GPS sites [51]. IJ05_R2 and W12a are regional GIA models constrained by extensive geological and glaciological data. Uncertainty in IJ05_R2 is estimated to be 1.40 mm/y according to the difference between the uplift rate derived from IJ05_R2 and that observed by six GPS stations with the observation period over 3000 days [52]. Uncertainty in W12a is estimated according to the difference between the provided upper and lower bounds of uplift rate. The three representative GIA models, derived from different Earth models and observations, show significant differences in spatial distribution compared with other GIA models, but are in good agreement with the GPS observations in Antarctica, which we assume are suitable for investigating the effect on BMB and BWSV results.

In order to ensure the consistency of the study period, we utilized sparse GPS observations to force the GIA predicated uplift rates. The 57 GPS sites used here are selected from the 118 GPS sites given by Sasgen et al. [54], based on the selection criteria that the observation period is consistent with our study period and the associated errors are smaller than the uplift rate. Figure 5a–c shows the predicated uplift rates predicted by ICE-6G, IJ05_R2, and W12a, as well as the uplift rate observed by 57 GPS sites and their uncertainties.

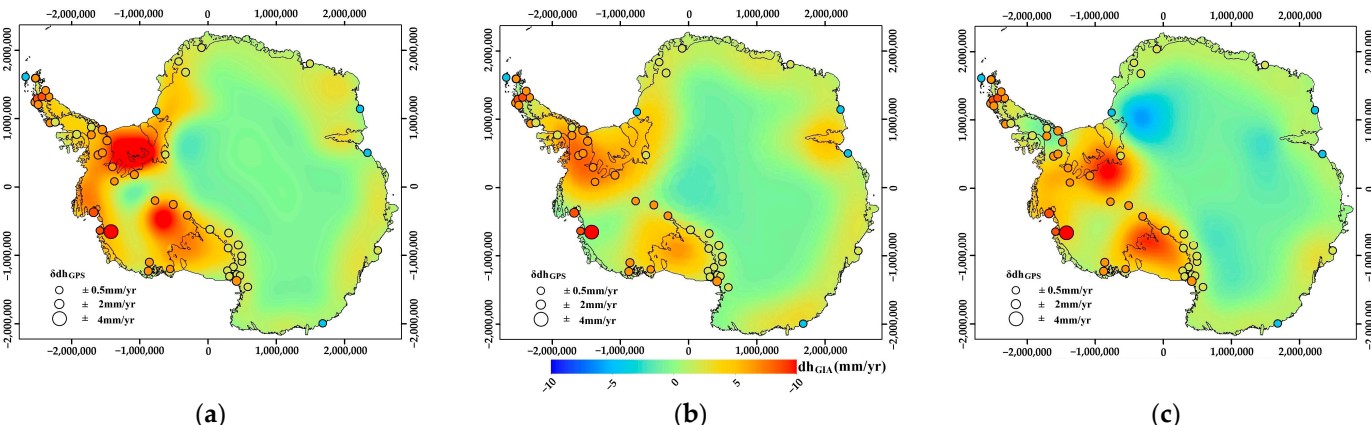

**Figure 5.** GIA predicated uplift rates from (**a**) ICE-6G, (**b**) IJ05_R2, (**c**) W12a, and uplift rates observed by 57 GPS sites and associated uncertainties during 2003–2009. The color of the circle donates the uplift rate and the radius donates the associated uncertainties.

### 2.4.4. Additional Datasets

Height variations caused by the spatio-temporal evolution of AIS firn layer is available from the Institute for Marine and Atmospheric Research Utrecht Firn Densification Model (IMAU-FDM) [34]. To simulate temporal evolution of density and height variations of the firn layer, the time-dependent IMAU-FDM is constrained by several datasets including surface mass balance, surface temperature, and wind speed from the regional atmospheric climate model RACMO2/ANT [34]. The time-dependent IMAU-FDM data period used in this study is consistent with the ICESat simultaneous observation period, and the linear height variations trends are subtracted through the least-square adjustment method in Section 2.4.1. Figure 6a,b show the linear height variations trend derived from IAMU-FDM over the study period and the associated uncertainties. For the consistency of spatial resolution, a 300 km Gaussian filter was also performed on IAMU-FDM derived surface height variations.

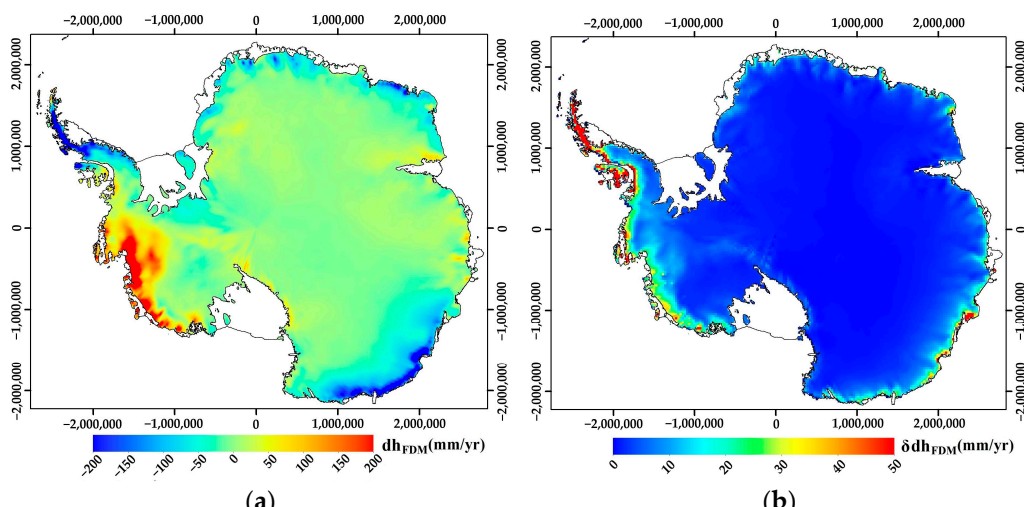

**Figure 6.** (**a**) Linear height variations trend derived from IAMU-FDM over the study period and (**b**) associated uncertainties.

Basal melting rate data used to identify basal melting/freezing and estimate BWSV are available from Pattyn [11], inferred through a hybrid method that combines prior information (such as on-site measurements, topography, accumulation, surface temperature, geothermal heat flow data) with the ice sheet/ice stream model. Although the period of the prior information (1980–2004, [11,55]) is different from that of this study, this period

discrepancy has little influence on the estimation of BWSV due to the stable basal conditions caused by the isolation of the overlying ice sheet. Figure 7a,b shows the basal melting rates over AIS and the associated uncertainties. For the consistency of spatial resolution, a 300 km Gaussian smoothing filter was also performed on the basal melting rate to match the spatial resolution of BMB.

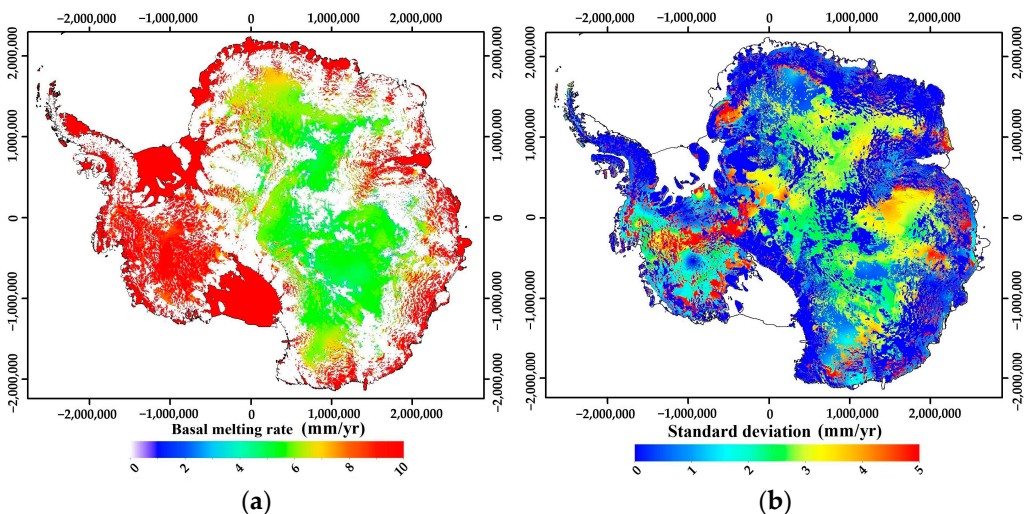

(**a**)  (**b**)

**Figure 7.** (**a**) Basal melting rates over AIS (truncated at 10 mm/y) and (**b**) associated standard deviation. This dataset is available from Pattyn [11].

## 3. Results and Discussion

### 3.1. Basal Mass Balance Beneath Antarctic Ice Sheet

Figure 8 displays the total BMB results for each iteration related to three GIA models. All BMB results converge to a negative value since the seventh iteration, showing the stability of the iteration method. Among them, total result of BMB (ICE-6G) is close to that of BMB (W12a) and about 5 Gt/yr larger than that of BMB (IJ06_R2), which indicates that using different GIA models has less influence on BMB results.

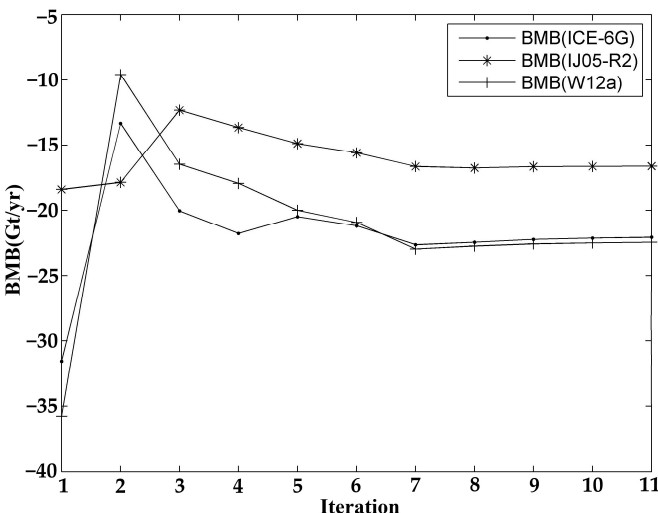

**Figure 8.** Total basal mass balance (BMB) results for each iteration over AIS.

Table 1 shows the three regional BMB rates in 18 drainage basins and associated standard deviations (Std). Figure 9 displays the comparison of regional BMB rates among the three results (Figure 9a), and error contributions of each input dataset (Figure 9b–d). The drainage basins division method used for regional BMB rates statistics comes from Rignot et al. [56] (spatial division is shown in Figure 9), and is employed in this study

based on the spatial similarity between simulated basal water pathways and observed ice flows. As shown in Table 1 and Figure 9a, regional BMB rates related to three GIA models are similar in most basins, while the large difference in B9 is primarily responsible for the discrepancy of total variation rates of the three BMB results. Among them, B5, B6, and B9 exhibit obvious basal mass increases (regional BMB rates lager than 10 Gt/yr and larger than associated Std), while B2, B3, and B11 display obvious basal mass decreases. Figure 9b–d displays consistency of the regional Std for the three BMB results, with the larger Std (greater than 5 Gt/yr) located in B3, B5, B8, B9, and B17, and the smaller Std located in B10, B12–B15, and B18, which is mainly determined by the area of the basins. Specifically, the main error sources in estimating BMB come from satellite gravimetry (35%, including GRACE + degree 1&2), ICB (26%), IMAU-FDM (14%), GIA (15%), accounting for 90% of the total Std of BMB, while errors of GPS (6%) and ICESat (4%) have a lesser effect on BMB. It is worth noting that errors generated by the iteration procedure itself are difficult to assess, while these errors are considered to be small due to the convergence of the iteration result and are therefore not contained in the uncertainties result. Overall, regional average BMB rates (average of regional BMB rates related to three GIA models) in the East Antarctic Ice Sheet (EAIS, including B1–B8, B17, and B18) and West Antarctic Ice Sheet (WAIS, including B9–B12 and B16) are $11 \pm 20$ Gt/yr and $-31 \pm 8$ Gt/yr, accounting for 23% and 30% of the corresponding documented ice-sheet mass balance [57], respectively. The regional average BMB rate in the Antarctic Peninsula Ice Sheet (APIS, including B13–B15) is very low ($-1 \pm 2$ Gt/yr). The total average BMB rate over AIS is $-21 \pm 22$ Gt/yr, accounting for 29% of the documented total ice-sheet mass balance rate ($-76 \pm 20$ Gt/yr, during 2003–2010) [57].

**Table 1.** Regional BMB rates in 18 drainage basins and associated standard deviations, in Gt/yr.

| Basin | BMB (ICE-6G) | | BMB (IJ05_R2) | | BMB (W12a) | | Basin | BMB (ICE-6G) | | BMB (IJ05_R2) | | BMB (W12a) | |
|-------|------|-----|------|-----|------|-----|-------|------|-----|------|-----|------|-----|
| | Rates | Std | Rates | Std | Rates | Std | | Rates | Std | Rates | Std | Rates | Std |
| B1 | 4 | 4 | 4 | 5 | 4 | 4 | B11 | −41 | 3 | −40 | 4 | −45 | 3 |
| B2 | −10 | 4 | −11 | 4 | −11 | 3 | B12 | −1 | 1 | −1 | 1 | −1 | 1 |
| B3 | −10 | 7 | −9 | 9 | −11 | 7 | B13 | 0 | 1 | 0 | 1 | −1 | 1 |
| B4 | −1 | 4 | −1 | 5 | −1 | 4 | B14 | 1 | 1 | 1 | 1 | 1 | 1 |
| B5 | 20 | 7 | 21 | 8 | 20 | 7 | B15 | −2 | 1 | −2 | 1 | −2 | 1 |
| B6 | 12 | 4 | 11 | 5 | 12 | 4 | B16 | 4 | 4 | 2 | 5 | 0 | 4 |
| B7 | 2 | 2 | 1 | 2 | 2 | 2 | B17 | 0 | 12 | 0 | 13 | 3 | 11 |
| B8 | −6 | 9 | −8 | 10 | −4 | 8 | B18 | 0 | 2 | 0 | 2 | 0 | 2 |
| B9 | 9 | 5 | 18 | 6 | 15 | 5 | | | | | | | |
| B10 | −4 | 1 | −4 | 1 | −4 | 1 | Total | −23 | 21 | −18 | 24 | −22 | 20 |

Figure 10 displays spatial distributions of BMB rates and associated standard deviations. Red colours in Figure 10a–c are basal mass increases regions with positive BMB rates, while blue colours are basal mass decrease regions with negative BMB rates; shadows represent unsignificant regions where the BMB rates are lower than the associated Std. As shown in Figure 10a–c, three BMB results show similar spatial distributions with the obvious basal mass changes occurring mainly in WAIS, marginal regions of EAIS, and Wilkes Land. The Std of the three BMB results (Figure 10d–f) also show identical spatial distributions: APIS and WAIS regions possess the largest Std ($\geq$15 mm/yr), mainly from uncertainties of satellite gravimetry (accounting for about 25% of total Std), ICB (~25%), IMAU-FDM (~20%) and GPS (~20%); medium Std (10–15 mm/yr) are located in the marginal region of EAIS, mainly from satellite gravimetry (~40%), ICB (~30%) and IMAU-FDM (~20%); while the low Std (<10 mm/yr) covers a large extent the interior of EAIS, which mainly comes from satellite gravimetry (45%) and ICB (35%).

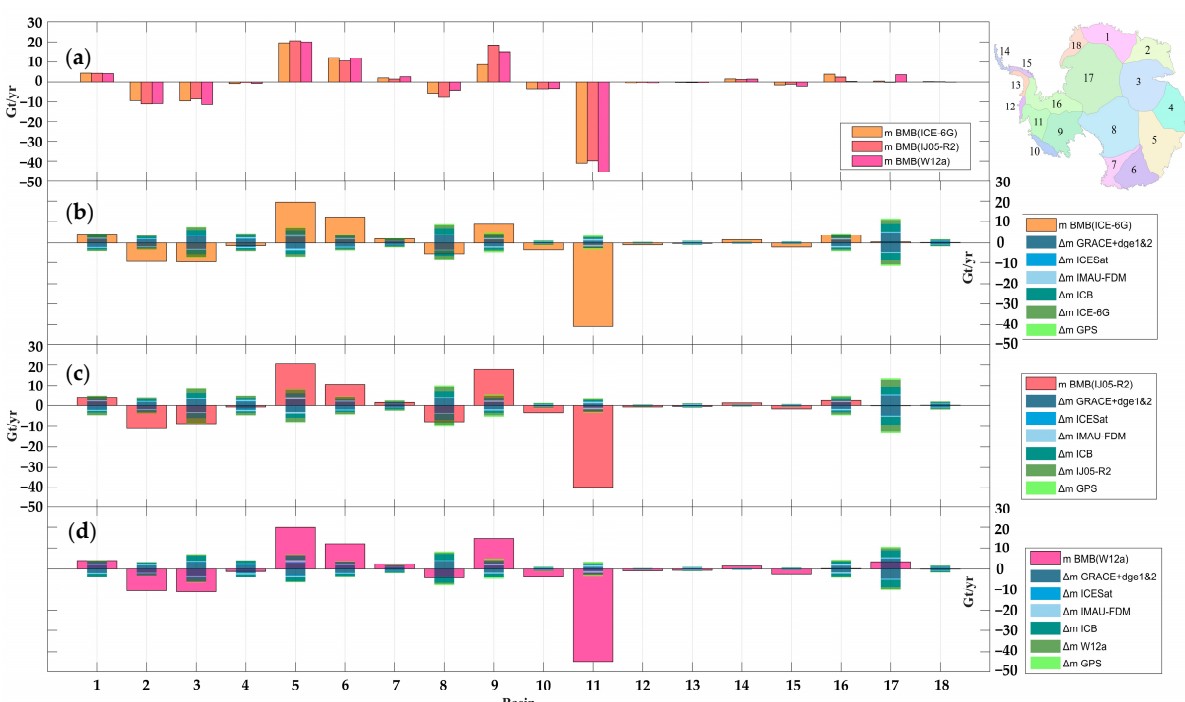

**Figure 9.** (**a**) Comparison of regional BMB rates in 18 drainage basins and (**b–d**) error contributions of each input datasets.

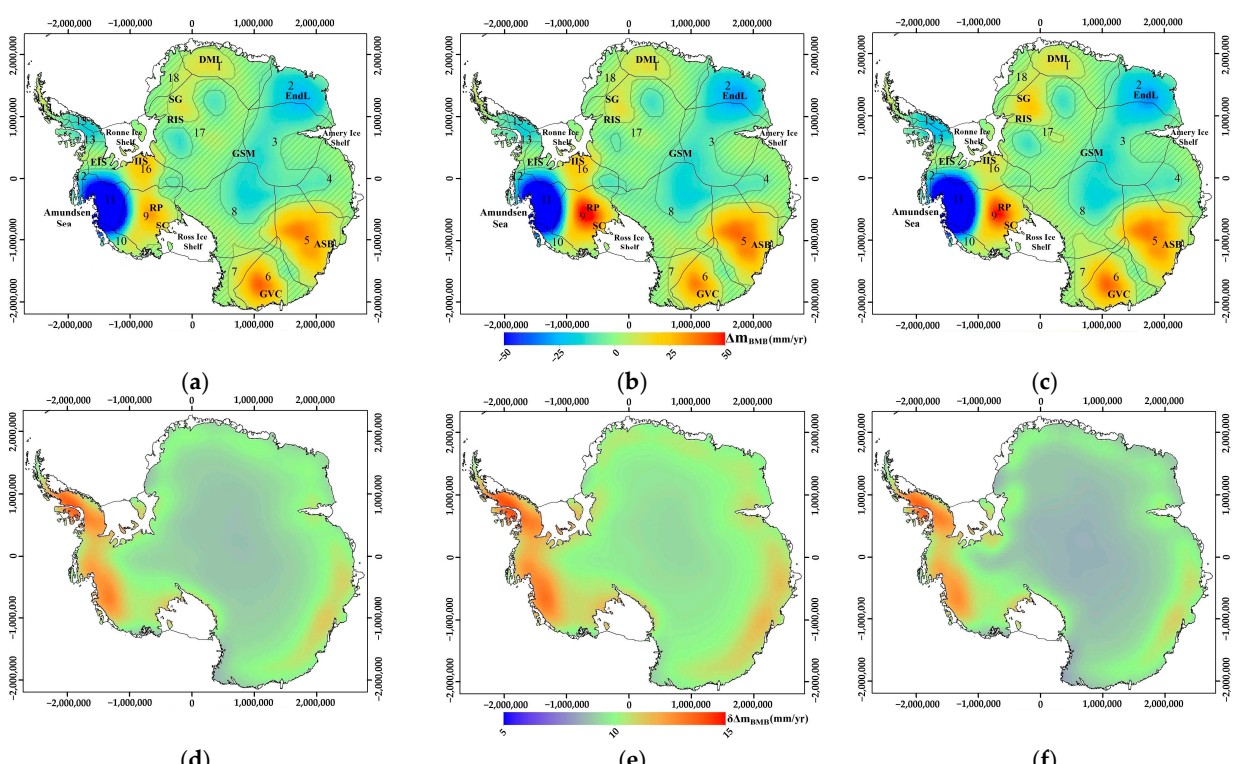

**Figure 10.** Spatial distribution of BMB related to (**a**) ICE-6G, (**b**) IJ05_R2 and (**c**) W12a in unit of mm/yr, and (**d–f**) associated standard deviations. Shadows are regions where the absolute values of BMB rates are lower than the associated standard deviations. SC = Siple Coast; DML = Dronning Maud Land; EndL = Enderby Land; ASB = Aurora Subglacial Basin; IIS = Institute Ice Stream; GVC = George V Coast; RIS = Recovery Ice Stream; SG = Slessor Glacier; RP = Rockefeller Plateau; GSM = Gamburtsev Subglacial Mountain.

Significant basal mass increases (with the positive BMB rates greater than the associated Std) occur mainly in the Rockefeller Plateau (RP), Siple Coast (SC), George V Coast (GVC), Aurora Subglacial Basin (ASB), Dronning Maud Land (DML), Slessor Glacier (SG), Recovery Ice Stream (RIS) and Institute Ice Stream (IIS), which is mainly attributed to the low basal hydrological potential in these regions that facilitating basal water accumulation (Figure S3, [58]). In some of these regions, the regional basal mass increases have been verified by other studies. For example, basal mass increases in the SC region have been demonstrated by revealing the basal water accumulation mechanism beneath the MacAyeal and Whillans ice streams [2,59]; basal mass increases in RIS region are proven to be caused by subglacial lake water discharging into the bedrock trench beneath the Recovery Glaciers [60]; furthermore, some studies have found that the basal water in Gamburtsev Subglacial Mountain (GSM) flows upward to basal ridges [61], and a similar pattern is also revealed in our BMB result, although not significant enough. On the other hand, significant basal mass decreases occur mainly along Amundsen Sea coast, in the interior of EAIS, and in the Enderby Land (EndL) region. Among these, basal mass decreases along the Amundsen Sea coast region are proven to be caused by the basal water, generated by basal geothermal flux-induced active ice melting [62] and discharged into Amundsen Sea through basal channels [59]; basal mass decreases in the interior of EAIS are due to the outward flow of the basal water driven by basal hydrological potential gradient [58]. However, the basal mass decreases in the EndL region lack verification and need further exploration.

### 3.2. Basal Water Storage Variations Beneath Antarctic Ice Sheet

Table 2 shows regional basal water storage variations (BWSV) rates and associated Std. Figure 11 displays the comparison among regional BWSV rates of the three results and associated error contributions of each input dataset. As shown in Figure 11a, regional BWSV rates related to three GIA models are also similar in most of the drainage basins: B5, B6 and B9 exhibit obvious basal water increases, B11 displays obvious basal water decreases, while other drainage basins show little basal water variation. Overall, regional average BWSV rates (average of regional BWSV rates related to three GIA models) in EAIS and WAIS are $47 \pm 21$ Gt/yr and $-4 \pm 8$ Gt/yr, respectively, while no obvious BWSV occurs in APIS regions. The total average BWSV rate over AIS is $43 \pm 23$ Gt/yr, which is 22 Gt/y lower than the basal meltwater increase rate (65 Gt/yr) [11], indicating that most of the increased basal meltwater is stored in the ice-bed interface.

**Table 2.** Regional BWSV rates in 18 drainage basins and associated standard deviations, in Gt/yr.

| Basin | BWSV (ICE-6G) | | BWSV (IJ05_R2) | | BWSV (W12a) | | Basin | BWSV (ICE-6G) | | BWSV (IJ05_R2) | | BWSV (W12a) | |
|---|---|---|---|---|---|---|---|---|---|---|---|---|---|
| | Rates | Std | Rates | Std | Rates | Std | | Rates | Std | Rates | Std | Rates | Std |
| B1 | 5 | 4 | 4 | 5 | 4 | 4 | B11 | −16 | 4 | −15 | 4 | −15 | 4 |
| B2 | −1 | 4 | −2 | 4 | −1 | 3 | B12 | 0 | 1 | 0 | 1 | 0 | 1 |
| B3 | −2 | 8 | −2 | 9 | −2 | 7 | B13 | 0 | 1 | 0 | 1 | 0 | 1 |
| B4 | 4 | 5 | 4 | 5 | 4 | 4 | B14 | 0 | 1 | 0 | 1 | 0 | 1 |
| B5 | 17 | 7 | 18 | 8 | 15 | 7 | B15 | 0 | 1 | 0 | 1 | 0 | 1 |
| B6 | 11 | 4 | 11 | 5 | 10 | 4 | B16 | 0 | 5 | 4 | 5 | 2 | 5 |
| B7 | 4 | 2 | 4 | 2 | 4 | 2 | B17 | 6 | 12 | 6 | 14 | 7 | 11 |
| B8 | 2 | 9 | 1 | 10 | 3 | 8 | B18 | 2 | 2 | 2 | 2 | 2 | 2 |
| B9 | 6 | 5 | 13 | 6 | 10 | 5 | | | | | | | |
| B10 | 0 | 1 | 0 | 1 | 0 | 1 | Total | 38 | 22 | 48 | 25 | 43 | 21 |

Figure 11b–d displays similar regional Std among the three BWSV results, with almost identical magnitude to the BMB results. The main error sources are also identical to that of BMB, while its rate drops to 80% (including: 32% from satellite gravimetry, 23% from ICB, 13% from IMAU-FDM, and 12% from GIA) of the total Std, due to the introduction of basal melting errors (accounts for 8% of the total Std of BWSV). Although the uncertainties of the result are derived from various input data, the total Std are relatively small, which is

attributed to: 1. the usage of the satellite gravimetry average, ICB correction average, and GIA model average reduces the uncertainties of the input datasets; 2. the block crossover analysis method used in satellite altimetry reduces the uncertainties of the observed height variations; 3. the usage of surface density discrimination method avoids introducing additional errors from regional atmospheric climate model.

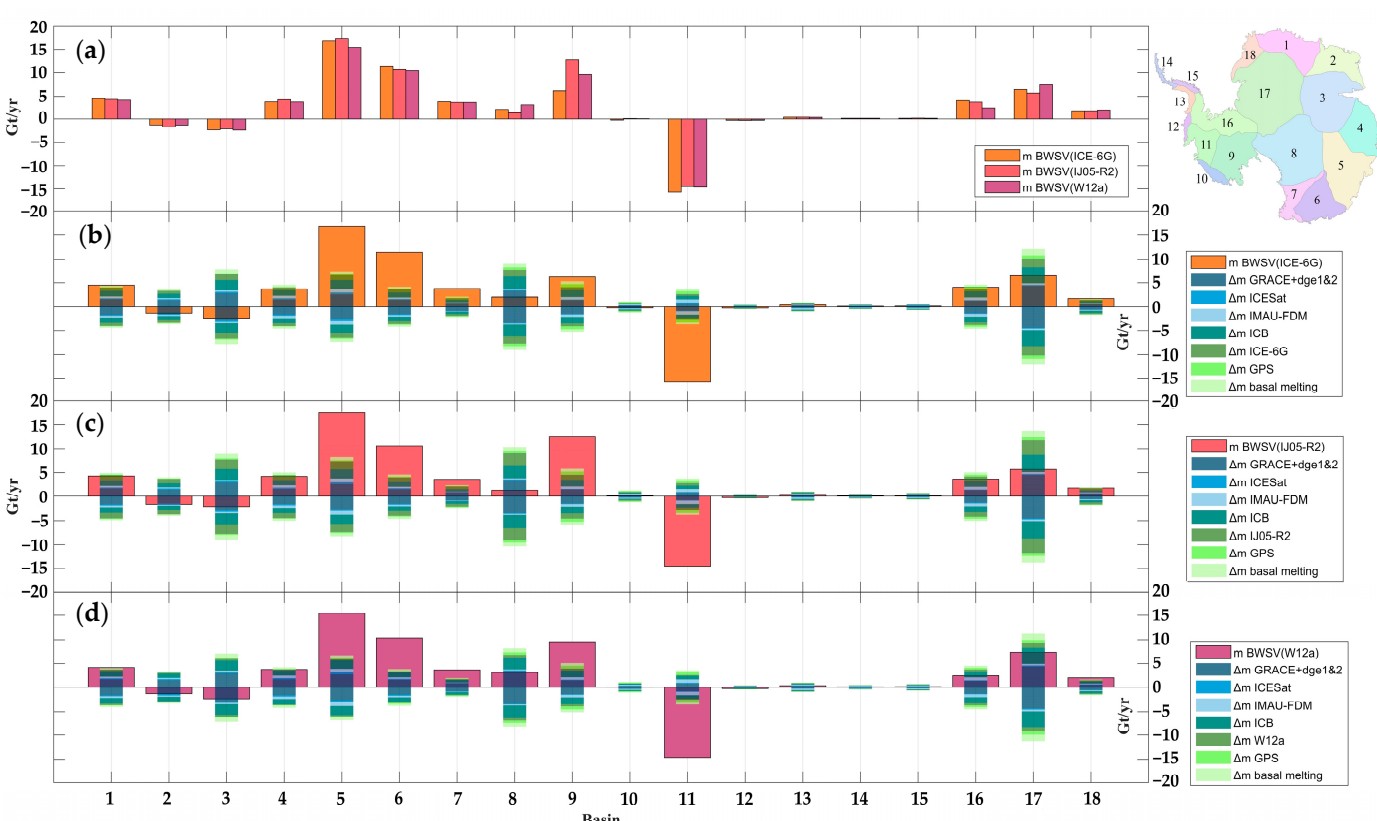

**Figure 11.** (**a**) Comparison of regional BWSV rates in 18 drainage basins, and (**b–d**) error contributions of input dataset to regional standard deviations of BWSV rates.

Figure 12 displays the spatial distributions of BWSV rates and associated uncertainties, as well as the location of subglacial lakes. Red colours in Figure 12a–c are regions with increased basal water storage that arise from basal water inflow and basal ice melting, while blue colours are regions with decreased basal water storage that are caused by basal water runoff. Blue dots in Figure 12a are locations of active subglacial lakes inferred from the surface height variations of AIS [63], grey dots in Figure 12b are locations of definite or fuzzy subglacial lakes detected by radio-echo sounding (RES) technique [64]. The spatial distributions of BWSV (Figure 12a–c) are similar to that of BMB, and differences are located mainly in marginal regions of EAIS where a more extensive basal water increases exists. In WAIS and marginal regions of EAIS (for example, IIS, RP, GVC, ASB, and SG regions in Figure 12a), the spatial distributions of increased basal water are consistent with those of active subglacial lakes, suggesting that basal water storage in most active subglacial lakes is increasing, despite the frequent water drainage events. These increased basal water storages are related to the following reasons: 1. regional low basal hydrological potential facilitates the convergence of surrounding basal water; 2. regional active melting of the bottom of ice layer contributes to the replenishment of basal water storage. The former reason for basal water storage increasing has also been supported by other studies, such as the continued basal water increases in most subglacial lakes in the RP region obtained from multi-mission satellite altimetry [65]. However, the latter reason for basal water storage remains unverified due to its limited contribution to height variations. Besides, definite or fuzzy lakes (grey circles in Figure 12b) are situated mainly in low-BWSV regions of

EAIS, which could be explained by lacking active basal water migration in these regions. However, exceptions are Concordia Subglacial lakes in Dome C (DC) region, where the increased basal water storage might be attributed to the fierce basal ice melting in Concordia Ridge, Concordia Subglacial Lakes, and Vincennes Basin [66], or regional 'hydrological depression'-induced long-term basal water accumulation.

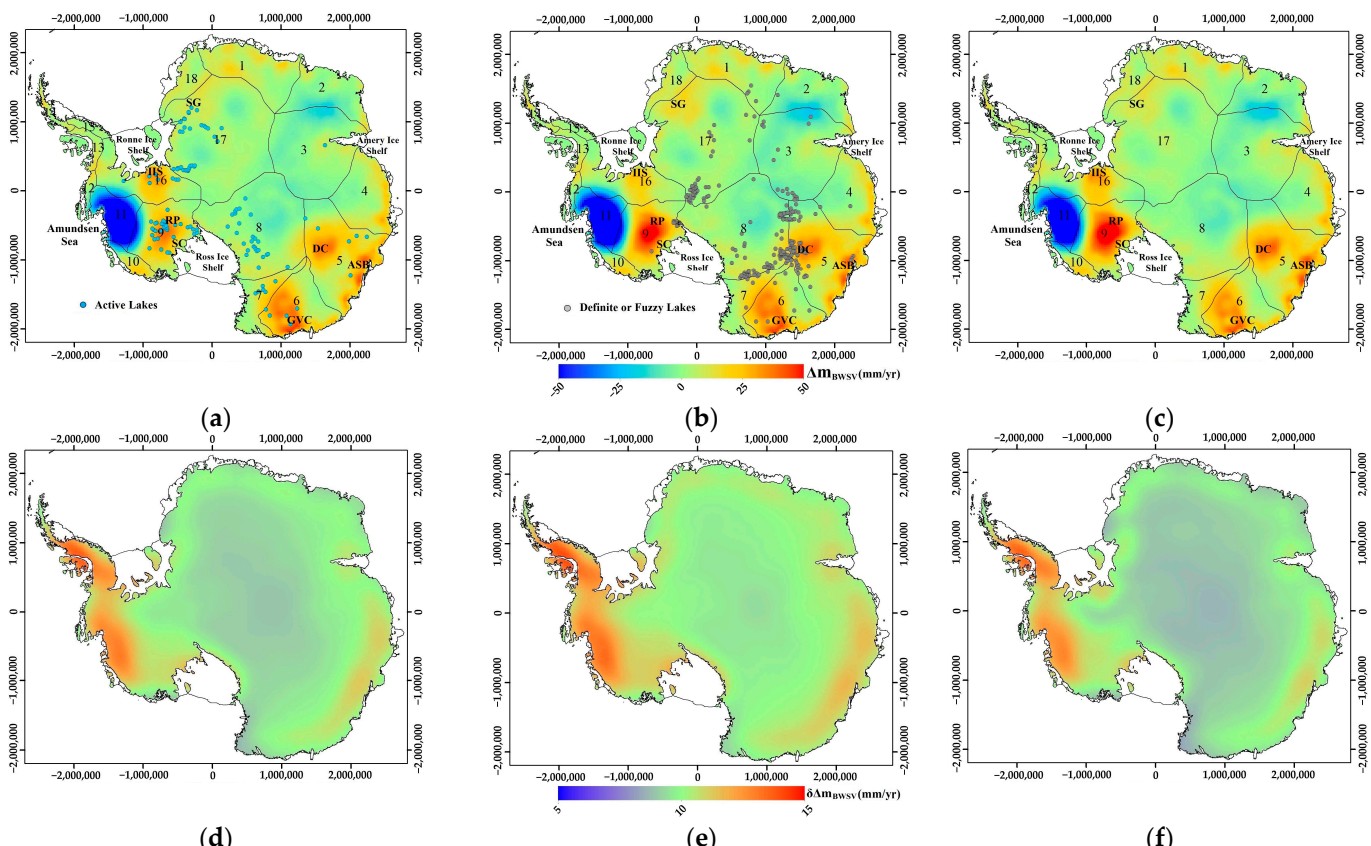

**Figure 12.** Spatial distribution of BWSV related to (**a**) ICE-6G, (**b**) IJ05_R2 and (**c**) W12a, and (**d**–**f**) associated standard deviations (in mm/yr). Blue and grey dots in (**a**,**b**) represent active subglacial lakes and definite or fuzzy subglacial lakes, respectively. DC = Dome C.

## 4. Conclusions

This study presents a layered gravity density forward/inversion method in combination with the iteration approach to estimate BMB and BWSV. The input datasets include multi-source satellite data and relevant models, most of which span from 2003 to 2009. Our results show that all the total BMB rates converge to negative values and display identical spatial distributions, which shows the potential and stability of detecting Antarctic BMB through the presented method.

The total BMB over AIS decreases at an average rate of $-21 \pm 22$ Gt/yr (EAIS: $11 \pm 20$ Gt/yr, WAIS: $-31 \pm 8$ Gt/yr, APIS: $-1 \pm 2$ Gt/yr), accounting for 29% of the mass balance rate ($-76 \pm 20$ Gt/yr) estimated by Shepherd et al. [57]. Spatially, obvious basal mass decreases are located mainly along Amundsen Sea coast, the interior of EAIS, and the Enderby Land region, while basal mass increases are situated mainly in the Rockefeller Plateau, Siple Coast, Institute Ice Stream regions, and the marginal of EAIS. The spatial distribution of BWSV was similar to that of BMB, with a rate of $43 \pm 23$ Gt/yr (EAIS: $47 \pm 21$ Gt/yr, WAIS: $-4 \pm 9$ Gt/yr, APIS: $0 \pm 1$ Gt/yr). In WAIS and marginal regions of EAIS, similar spatial distribution between increased basal water and active subglacial lakes suggested that the water storage in most active subglacial lakes is increasing, despite the frequent water drainage events. Basal water storage in most regions with definite or fuzzy lakes is relatively

stable, with exceptions in Concordia Subglacial lakes regions, where the increased basal water storage caused by fierce basal ice melting and long-term basal water accumulation.

Major error sources in estimating BMB and BWSV come from satellite gravimetry errors (including GRACE and coefficients of degree 1&2) and ICB correction errors in satellite altimetry, which account for 55% of the total errors. Therefore, the errors in BMB and BWSV results are expected to decrease substantially, provided the progress of the harmonization of the benchmarks for different satellite observations continues.

In summary, the method presented in this paper can be used to calculate the Antarctic continental BMB and BWSV, based on existing satellite observation data and several relevant models. The results could contribute to the understanding of detailed mass variations of AIS and the changes in basal heat flux, basal effective stress, and ice dynamics in Antarctica.

**Supplementary Materials:** The supporting information for gravity forward modelling method, the layered gravity density inversion method, as well as basal hydraulic potential of AIS can be downloaded at: https://www.mdpi.com/article/10.3390/rs14102337/s1. C.f., [67,68].

**Author Contributions:** Conceptualization, J.K., Y.L. (Yang Lu) and H.S.; methodology, J.K. and Y.L. (Yang Lu); validation, J.K., Y.L. (Yan Li) and Z.Z.; formal analysis, J.K., Y.L. (Yang Lu) and Y.L. (Yan Li); investigation, Y.L. (Yan Li) and H.S.; resources, J.K., Y.L. (Yang Lu) and H.S.; data curation, J.K. and H.S.; writing—original draft preparation, J.K.; writing—review and editing, Y.L. (Yang Lu), Y.L. (Yan Li) and Z.Z.; visualization, J.K. and Y.L. (Yang Lu); supervision, Y.L. (Yang Lu) and Z.Z.; project administration, Y.L. (Yang Lu) and H.S.; funding acquisition, Y.L. (Yang Lu) and H.S.; All authors have read and agreed to the published version of the manuscript.

**Funding:** This research was jointly funded by the National Natural Science Foundation of China (Grant No. 41674085, 41874093, and 42074094), Independent project of State Key Laboratory of Geodesy and Earth's Dynamics (S21L6401).

**Data Availability Statement:** BMB and BWSV data are available at https://github.com/Kangjingyu17/BMB-BWSV.git. Dataset used in this study is listed GRACE: ftp://isdcftp.gfz-potsdam.de/grace/Level-2/; ICESat: https://nsidc.org/data/icesat/; BEDMAP2: https://secure.antarctica.ac.uk/data/bedmap2/; GPS: https://dep1doc.gfz-potsdam.de/documents/102; ICE-6G: https://www.atmosp.physics.utoronto.ca/~peltier/data.php; W12a: http://www.pippawhitehouse.com/; all accessed dates are 29 March 2022. Any other specific data of this study are available on request to the authors.

**Acknowledgments:** The author would like to thank Regional glacial isostatic adjustment and CryoSat elevation rate corrections in Antarctica (REGINA), Peltier, and Whitehouse for the distribution of GPS, ICE-6G, and W12a data respectively. We are also grateful to Ligtenberg, Pattyn, Van Liefferinge, and Ivins, for the use of FDM/firn density data, basal melting rate data, and IJ05_R2 data.

**Conflicts of Interest:** The authors declare no conflict of interest.

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
