# Peer review of "Antarctic Basal Water Storage Variation Inferred from Multi-Source Satellite Observation and Relevant Models"

_remotesensing, doi:10.3390/rs14102337_

Round 1

Reviewer 1 Report

Review for Kang et al.,

This study proposed a layered gravity density forward/inversion iteration method to calculate Antarctic Basal water storage. The reseach designs well and the method are adequately described. So I would recommend to be accept after minor revision.

Some comments or suggestions:

  1. Three GIA models are used to calculate the Earth’s isostatic adjustment. However, as far as I know, there are several more GIA models, such as Geruo13, Pailson07 and Wang's. In this study, average results of the three GIA models are used to reduce the uncertainties. So I would suggestion to utilize more GIA models rather than the selected three. Or, the author can present the priorities of the three models.
  2. The references of subglacial lakes seem to be out of date. Please track the newest research on subglacial lakes.
  3. The Conclusion and discussions section is more like a conclusion. I would suggest to separate the two parts and focus more on the discussion.

Reviewer 2 Report

Comments and Suggestions for Authors:

The paper titled “Antarctic Basal Water Storage Variations Inferred from Multi-source Satellite Observation and Relevant Models” deals with the estimation of Basal Mass Balance from combined satellite geodetic data. The results of that estimation were finally further used in combination with basal melting data, in order to estimate Antarctic basal water storage variations.

The subject of the present research is very interesting. The manuscript is well structured and comprehensive. The research is properly organized and conducted. What the authors want to say in the present study is well said and stated and they mention if any similar studies have been carried out previously. However, the authors do not make clear what the need of carrying out this study was. Usually, at the end of the introduction the authors should explain the aim of the study. At the present manuscript, they just make a brief description of the paper’s content. Is this necessary? In my opinion, it is more necessary to explain the objectives of the present research, as well as its novelty aspects. The authors should explain them more clearly. This is important for every scientific paper, so please consider and revise.

Although the subject of the paper is very interesting, there is a difficulty by the first impression of the manuscript. More particularly, the extent of the “3.Results” and “4.Conclusion and discussions” sections all together is the half of the “2.Materials and Methods” section alone. Meaning that, the “Materials and Methods” part is very long, while the following parts were the results and conclusions are figured, are way too limited. Hence, the “Results” part could be enhanced and improved i.e. with the presentation of data in diagrams in order the manuscript becomes more interesting to the reader. Furthermore, regarding the structure of the manuscript, usually, “Results” and “Discussion” go together as a section, and “Conlcusions” is the last one, where the most important findings of the study should be referred. But in the present manuscript,”Conclusion and Discussions” are presented together. Is this right? The authors should consider and revise these parts, which should be enhanced and improved, in order the manuscript becomes more appropriate and meets the requirements for publication in such a significant international journal like “Remote Sensing”.

Regarding the English language style, although I do not feel qualified to judge about it, it probably requires some editing and needs to be improved throughout the manuscript.

As a suggestion that, in my opinion, would make the manuscript more interesting, maybe the authors should consider the possibility of the graphical presentation of the results. There is a lot of numerical information and data obtained by the research. Apart from depicted in variation maps, they are also referred in tables and in the text. Probably it would be much more informative if the results were also demonstrated in diagrams. That way, the manuscript would become much more comprehensive to the reader.

So these are some improvements that could be made throughout the manuscript. They are the only reason why my recommendation is to accept the manuscript after major revision and not in its present form. It would be preferable if the authors would take them into account

Reviewer 3 Report

see the attachment

Round 2

Reviewer 2 Report

The revised manuscript was carefully checked, as well as the cover letter with the authors' responses. The manuscript has been significantly updated. It is obvious that the authors made effort in order to meet the reviewer’s suggestions and most of them were taken into account in the best way!

Hence, by my point of view, the revised manuscript is more than welcome to  be accepted in its present form.